# Human MAIT cells respond to and suppress HIV-1

Chansavath Phetsouphanh[1,2]*[†], Prabhjeet Phalora[1†], Carl-Philipp Hackstein[1†], John Thornhill[3], C Mee Ling Munier[2], Jodi Meyerowitz[1], Lyle Murray[1], Cloete VanVuuren[4], Dominique Goedhals[5], Linnea Drexhage[6], Rebecca A Russell[6], Quentin J Sattentau[6], Jeffrey YW Mak[7,8], David P Fairlie[7,8], Sarah Fidler[3], Anthony D Kelleher[2], John Frater[1], Paul Klenerman[1]*

[1]Peter Medawar Building for Pathogen Research, University of Oxford, Oxford, United Kingdom; [2]The Kirby Institute, University of New South Wales, Sydney, Australia; [3]Imperial College London, London, United Kingdom; [4]Military Hospital, Bloemfontein, South Africa; [5]Division of Virology, University of the Free State/ National Health Laboratory Service, Free State, South Africa; [6]Sir William Dunn School of Pathology, University of Oxford, Oxford, United Kingdom; [7]ARC Centre of Excellence in Advanced Molecular Imaging, Institute for Molecular Bioscience, The University of Queensland, Brisbane, Australia; [8]ARC Centre of Excellence in Advanced Molecular Imaging, Institute for Molecular Bioscience, The University of Queensland, Brisbane, Australia

**Abstract** Human MAIT cells sit at the interface between innate and adaptive immunity, are polyfunctional and are capable of killing pathogen infected cells via recognition of the Class IB molecule MR1. MAIT cells have recently been shown to possess an antiviral protective role in vivo and we therefore sought to explore this in relation to HIV-1 infection. There was marked activation of MAIT cells in vivo in HIV-1-infected individuals, which decreased following ART. Stimulation of THP1 monocytes with R5 tropic HIV_{BAL} potently activated MAIT cells in vitro. This activation was dependent on IL-12 and IL-18 but was independent of the TCR. Upon activation, MAIT cells were able to upregulate granzyme B, IFNγ and HIV-1 restriction factors CCL3, 4, and 5. Restriction factors produced by MAIT cells inhibited HIV-1 infection of primary PBMCs and immortalized target cells in vitro. These data reveal MAIT cells to be an additional T cell population responding to HIV-1, with a potentially important role in controlling viral replication at mucosal sites.

**\*For correspondence:**
c.phetsouphanh@gmail.com (CP);
paul.klenerman@ndm.ox.ac.uk (PK)

[†]These authors contributed equally to this work

**Competing interest:** The authors declare that no competing interests exist.

## Introduction

Mucosal-associated invariant T-cells (MAIT cells) are innate-like T cells that rapidly produce cytokines upon activation and express a semi-invariant T-cell antigen receptor (TCR) (*Birkinshaw et al., 2014*; *Zinser et al., 2018*). Human MAIT cells are typically defined by expression of the Vα7.2 TCR (rearranged in-combination with Jα33) in combination with phenotypic markers, including high levels of the C-type lectin CD161 and IL18R (*Kurioka et al., 2017*). MAIT cells are restricted by the evolutionary conserved non-polymorphic MHC-related protein MR1, which presents microbially derived vitamin B metabolites (*Birkinshaw et al., 2014*; *Eckle et al., 2015*; *Kjer-Nielsen et al., 2018*). Recognition of MR1-bound ligands from riboflavin synthesizing bacteria by the MAIT TCR leads to release of cytokines such as IFNγ, TNFα, and IL-17, as well as triggering their cytolytic function (*Kurioka et al., 2017*; *Kurioka et al., 2015*; *Leeansyah et al., 2015*).

MAIT cells have been extensively studied in the context of HIV-1 infection. Early and non-reversible loss of CD161++/MAIT cell frequencies has been observed in HIV-1 infection (*Salou et al., 2017*;

*Cosgrove et al., 2013*), and this loss has been confirmed in several other human studies (*Juno et al., 2019a*, *Sortino, 2018*; *Spaan et al., 2016*; *Khaitan et al., 2016*; *Fernandez et al., 2015*; *Eberhard et al., 2014*), as well as SIV infection of rhesus macaques (*Vinton et al., 2016*; *Juno et al., 2019b*). Depletion of MAIT cells in peripheral blood during HIV-1 infection may be caused by several factors. Firstly, down-regulation of CD161 expression may lead to an underestimation of CD161++ Vα7.2+ MAIT cells in blood. However, use of MR1/5-OP-RU tetramers and qPCR for the specific TCR has independently confirmed previous findings that MAIT cells are indeed depleted in the blood during HIV-1 infection (*Fernandez et al., 2015*; *Ussher et al., 2018*). Secondly, up-regulation of tissue homing markers α4β7 (*Juno et al., 2019b*) and chemokine receptors (CXCR6+, CCR2+, CCR5+, CCR6+, and CCR9+) and the identification of MAIT cells in affected tissues demonstrate that they can migrate into tissues during infection (*D'Souza et al., 2018*), although recovery of MAIT cell numbers in blood is not reproducibly seen upon viral suppression with Antiretroviral Therapy (ART). Increased bacterial translocation from the gut during HIV-1 infection and MAIT cell migration into these sites may lead to activation-induced cell death following activation via their TCR (*Cosgrove et al., 2013*).

It has been shown that MAIT cells have the ability to sense viral infections through specific cytokine-driven mechanisms, including IL-12, IL-15, IL-18, Type I interferons (*van Wilgenburg et al., 2016*; *Ussher et al., 2014*) and most recently TNF (*Provine et al., 2021*). These mechanisms have been defined in vitro and activation of MAIT cells in response to acute and persistent virus infections (Dengue, HCV, and Influenza) and vaccines has been clearly demonstrated in vivo in humans (*van Wilgenburg et al., 2018*; *van Wilgenburg et al., 2016*; *Loh et al., 2016*; *Provine et al., 2021*). Importantly, such activation is associated with protection against death in a lethal influenza challenge model in mice (*van Wilgenburg et al., 2018*), which provides proof-of-principle that such TCR-independent activity has an important biological role. While previous experiments showed that HIV-1 infection leads to a decrease of MAIT cells (*Cosgrove et al., 2013*), it remains to be determined if MAIT cells display any kind of antiviral activity in the context of HIV-1 infection.

Here, we investigated to what extent and by which mechanisms HIV-1 could activate MAIT cells and whether this resulted in measurable anti-HIV-1 activity. We assessed MAIT cells in the peripheral blood of HIV-1+ subjects during different stages of infection and, given the importance of the gastro-intestinal (GI) tract in HIV-1 pathogenesis, in rectal and ileal tissue samples to define activation and redistribution. We established a model using HIV$_{BAL}$ to activate MAIT cells in vitro, showing that this process is TCR independent but dependent on IL-12 and IL-18 stimulation. Upon activation, MAIT cells were able to upregulate HIV-1 restriction factors and reduce levels of infection. These data, taken with emerging data from the field, suggest that MAIT cells should be included in the repertoire of antiviral populations activated during HIV-1 infection, with a potentially important role for control at mucosal sites.

## Results
### MAIT cells are activated by HIV-1 in vivo

To first address MAIT cell activation during HIV-1 infection, PD-1, granzyme B (GzmB), and TIM-3 protein levels were measured pre- and post-ART in donors with Primary and Chronic HIV-1 infection (PHI and CHI respectively) (*Figure 1A and B*). Expression of all three molecules was very low in healthy controls ex vivo (median: PD-1 = 0.85%, GzmB = 0%, and TIM-3 = 0.15%). Moderate levels of GzmB+ MAIT cells were detected during PHI at baseline (10.06%), which significantly decreased after ART (2.27%, p < 0.05). Granzyme B expression on MAIT cells was even higher in CHI donors, with a median of 55% (IQR: 41.2–93.9%) of cells being positive; however, these levels dramatically decreased following 1 year of ART (22.45%, p < 0.001). Activation-induced inhibitory receptors PD-1 and TIM-3 followed a similar pattern of expression to granzyme B during acute and chronic HIV-1 infection. There was a ~ 2.7-fold decrease in PD-1 and ~2.3-fold decrease of TIM-3 expressing MAIT cells after ART in PHI donors (p < 0.05 and p < 0.01, respectively). This was also observed in CHI, whereby lower percentages of PD-1 and TIM-3 in MAIT cells were detected post-ART (~6.4-fold [p < 0.01] and ~10.8 fold [p < 0.01], respectively). TIM-3 was also expressed at elevated levels in Elite and Viraemic controllers (EC and VC respectively), compared to Healthy controls (HC) (*Figure 1—figure supplement 1*), indicating some ongoing activation even with low levels of virus (*Figure 1—source data 1*).

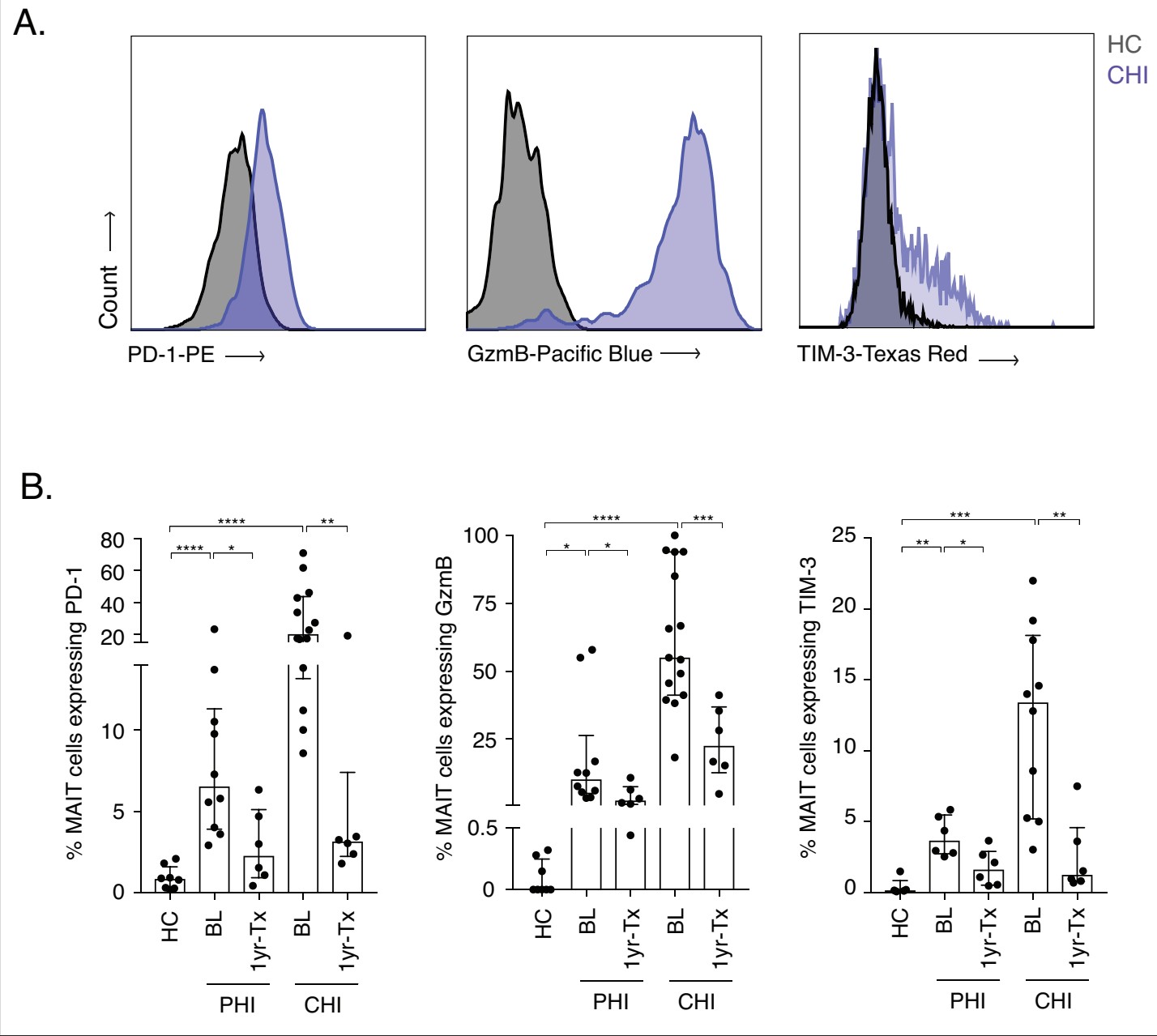

**Figure 1.** Increased activation and inhibitory marker expression on MAIT cells during HIV-1 infection. (**A**) Representative histograms showing upregulation of the activation/inhibitory markers PD-1, Granzyme B (GzmB) and TIM-3 in MAIT cells in chronic HIV-1 infection (CHI) compared to a healthy control (HC). (**B**) Increased expression of PD-1, GzmB, and TIM-3 on CD8+ CD161++ and Vα7.2+ MAIT cells during PHI and CHI chronic at baseline (BL) and 1 year post-ART (1 yr-Tx). Data points are biological replicates, shown as mean and standard deviation. *p < 0.05, **p < 0.01, *** p < 0.001, **** p < 0.0001; two-tailed t-tests.

The online version of this article includes the following source data and figure supplement(s) for figure 1:

**Source data 1.** Marker expression on MAIT cells during HIV-1 infection.

**Figure supplement 1.** TIM-3 expression on MAIT cells in LTNP.

To further assess the impact of HIV-1-induced activation on MAIT cells we tracked cell frequencies in blood in specific patient groups. While MR1-Tetramers loaded with the MAIT cell ligand 5-OP-RU represent the most reliable way to identify MAIT cells in the blood, they can also be identified as CD161++ Vα7.2+ cells, especially within the CD8+ population (***Kurioka et al., 2017***). In order to test whether this would also be the case in the gut, we co-stained CD161 and Vα7.2 with 5-OP-RU-loaded

or control MR1-tetramers and assessed whether CD161++ Vα7.2+ cells stained positive for the tetramers (*Figure 2—figure supplement 1*). Within the CD8+ and double-negative (DN) populations, almost all CD161++ Vα7.2+ cells reacted with the 5-OP-RU-loaded tetramer but not the 6FP-loaded control tetramer, suggesting that the combination of high CD161 expression with Vα7.2 can be used to identify MAIT cells within the CD8 and DN T-cell populations in the gut. CD161++ Vα7.2+ expression on CD8+ T cells was comparable with the MR1-5-OP-RU+ tetramer and could identify the majority of MAIT cells within this compartment.

As previously noted, loss of CD8+ MAIT cells was observed in HIV-1+ cohorts regardless of disease stage when compared to uninfected controls (*Figure 2A&B*). Looking at specific clinically defined populations, MAIT cell percentages were also very low in Long-Term Non-Progressors (LNTP), with a ~ 9 fold decrease in Elite controllers (ECs) and a ~ 17-fold decrease in Viraemic controllers (VCs) when compared to healthy controls (*Figure 2B*). Interestingly, a higher percentage of MAITs was observed in EC compared to those with chronic HIV-1 infection (median of 0.76% and 0.5% respectively, $p < 0.05$) indicating a potential relationship between MAIT cell frequency and divergent clinical outcomes. No recovery of MAIT cells in blood was observed in either CHI or PHI groups 1 year post-ART (*Figure 2C&D*). This observation was also consistent during long-term ART (up to 5 years), where MAIT cell percentages did not increase from 1 year post-ART (*Figure 2E*). Analysis of rectal vs blood derived MAIT frequencies in ART-treated patients revealed a clear positive correlation between the two compartments (rho = 0.69, $p < 0.05$) (*Figure 2F*), with relative enrichment of the cells in rectal tissue (*Figure 2G*). This indicates that both compartments are impacted in parallel and the decline in frequency in blood is unlikely to be accounted for by redistribution to the gut. These data taken together confirm and extend existing studies, indicating that MAIT cells are strongly activated in vivo by HIV-1 infection, varying according to levels of viral replication (*Figure 2—source data 1*).

## HIV-1 activates MAIT cells in vitro

We next addressed whether we could model HIV-1 activation of MAIT cells using an in vitro system to further define the mechanism of activation, which can vary between viral systems (*Provine et al., 2021*; *van Wilgenburg et al., 2018*), and which has not been previously demonstrated. THP-1 cells, a monocytic cell line, were infected with R5 tropic lab strain HIV_BAL for 6 hr and then incubated overnight with enriched CD8+ T cells from peripheral blood from healthy donors. IFNγ production from the CD161++ Va7.2+ population was measured to determine MAIT cell activation and formaldehyde fixed *E. coli* was included as a positive control (*Figure 3A*).

In this system, HIV_BAL was able to activate MAIT cells to similar levels as *E. coli* (15.98% and 19.65%; respectively) (*Figure 3A*), whereas an inactivated HIV_JRFL primary viral isolate did not have this effect (*Figure 3—figure supplement 1*). Since IL-12 and IL-18 in combination have been shown to be important in TCR-independent triggering of MAIT cells by some viruses such as Dengue (*van Wilgenburg et al., 2018 van Wilgenburg et al., 2016*), we sought to analyze this in our system. Blocking antibodies to IL-12 and IL-18 inhibited MAIT cell activation with *E. coli*, as has been previously shown (*Ussher et al., 2014*; *Kurioka et al., 2018*), and importantly also blocked HIV_BAL-mediated activation of MAIT cells (*Figure 3A*). This shows that HIV-1 can activate MAIT cells, similar to other viruses, in an IL12- and IL18-dependent manner.

## MAIT cells possess antiviral activity against HIV-1

Since HIV_BAL was able to activate MAIT cells in vitro, we sought to determine if viral stimulation of MAIT cells had an impact on HIV-1 infection. To that end, we set up an in vitro infection model, utilizing the CEM-GXR-GFP reporter cell line. Supernatants from unstimulated and IL-12/18 stimulated CD8+ enriched T cells were added to the reporter cells for 6 hr, followed by infection with HIV_BaL and GFP expression was measured 4 days post infection (*Figure 3B*). GFP expression was highest in the positive control samples containing only HIV_BAL (6.57%, $p < 0.01$) or HIV_BAL plus the unstimulated supernatant (6.35%). However, this was significantly reduced when stimulated supernatant was added (2.75%, $p < 0.01$). This inhibition was also observed when a HIV_JRFL-GFP virus was used instead (*Figure 3C*). The suppressive effect of MAIT cell-derived supernatants was evident irrespective of whether supernatants were derived from stimulated whole CD8s (*Figure 3C*, left) or from pure sorted MAIT cells (*Figure 3C*, right) or of the target cells used in the assay. The inhibition by the stimulated

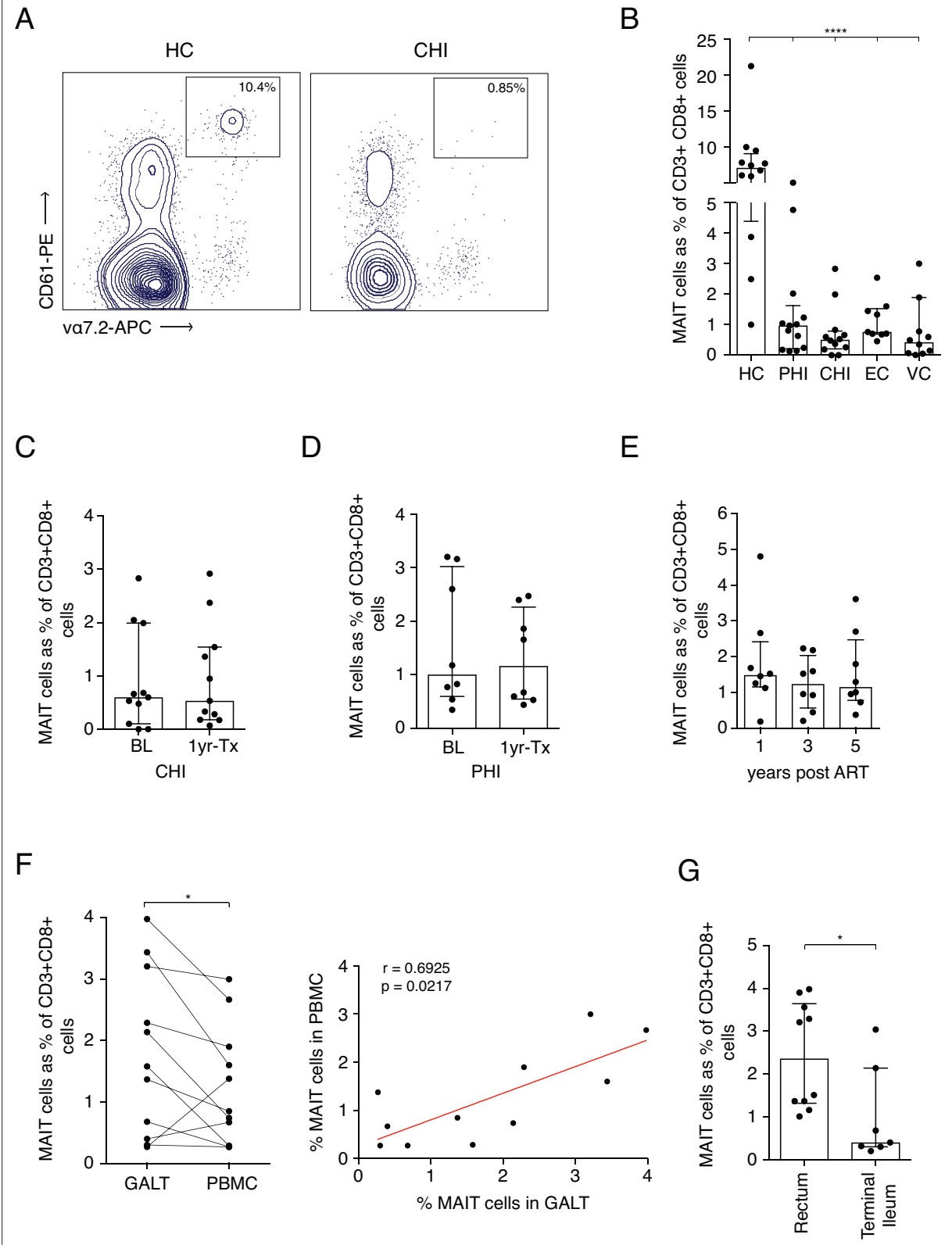

**Figure 2.** Frequency of MAITs cells in blood and intestine during HIV-1 infection. (**A**) Representative dot-plot showing loss of CD8+ MAIT cells, gated on CD161++ and Vα7.2+, in CHI compared to HC. (**B**) Loss of MAIT cells in peripheral blood in HIV-1+ donors at different HIV-1 stages PHI, CHI, EC (Elite Controllers), and VC (Viraemic controllers). (**C**) No recovery of MAIT cells post-ART in CHI. (**D**) No recovery of MAIT cells post-ART in PHI. (**E**) No recovery of MAIT cells following long-term ART. (**F**) Higher percentage of MAIT cells in rectal and illeal tissue compared to blood in matched PHI-treated donors.

*Figure 2 continued on next page*

*Figure 2 continued*

(**G**) MAIT cell percentages in the rectum compared to terminal ileum of PHI-treated donors. Data points are biological replicates, shown as mean and standard deviation. Spearman's correlation was used to calculate rho and p value. *p < 0.05, **p < 0.01, *** p < 0.001, **** p < 0.0001; two-tailed t-tests.

The online version of this article includes the following source data and figure supplement(s) for figure 2:

**Source data 1.** Frequency of MAITs cells in blood and tissue during HIV-1 infection.

**Figure supplement 1.** CD161++ Vα7.2+ identifies MAIT cells in tissue- comparable to MR1-5OPRU tetramers.

supernatants was also clearly titratable (***Figure 3—figure supplement 2***). Thus, IL12/18 stimulated MAIT cells are able to inhibit HIV-1 infection in vitro across a range of different reporter systems.

## Activated MAIT cells secrete effector molecules

Having observed that activated MAIT cells can exert an antiviral effect on $HIV_{BAL}$, we next sought to determine which effector molecules were important for mediating this effect. IL-12/18-stimulated PBMCs from healthy donors showed increased expression of granzyme B (198-fold), IFN-γ(57-fold)and Caspase 3 (9-fold) in the MAIT cell population (CD8+ CD161++ Vα7.2+) (***Figure 3—figure supplement 3***). To first assess whether IFN-γ contributes to the anti-HIV-1 phenotype, Jurkat-tat-R5 cells were infected with $HIV_{BAL}$ in the presence of IFNγ blocking antibodies and levels of p24 was measured. However, there was no impact of the blocking antibodies on infection levels indicating that IFN-γ is not required for the inhibitory phenotype in our model system (***Figure 3—figure supplement 4***). Next, as we observed high GzmB-expression in MAITs from HIV-1-infected individuals (***Figure 1***) and in vitro upon stimulation (***Figure 3—figure supplement 3***) we sought to determine whether viral inhibition was dependent upon cell contact. However, CEM-GXR cells either co-cultured with MAIT cells or separated in transwell plates showed no suppression of HIV-1 (***Figure 3—figure supplement 5***) suggesting that direct cell contact is not necessary for the MAIT-derived inhibition of HIV-1 infection (***Figure 3—source data 1***).

## Antiviral chemokines are secreted by MAIT cells in response to IL12/18 stimulation

Interestingly, IL-12/18-stimulated MAIT cells also upregulated production of the antiviral chemokine MIP-1β (CCL4) (***Figure 3—figure supplement 3***). This chemokine is a ligand for the HIV-1 entry receptor CCR5 and hence can restrict HIV-1 infection by blocking viral entry. Expression of CCL4 indicated a potential novel antiviral function for MAIT cells. To examine this further, sorted MAIT cells were stimulated with IL-12/18 overnight and supernatants were analyzed using an ELISA to measure CCL4 (***Figure 4A***) or a cytometric bead array (CBA) to measure CCL3 (MIP1α) and CCL5 (RANTES) (***Figure 4B***), the other two known ligands for the CCR5 receptor. Higher levels of all three chemokines were expressed by stimulated MAIT cells compared to the unstimulated controls (CCL4 p = *0.125*, CCL3 p = *0.0156*, CCL5 p = *0.0078*), although these differences were not statistically significant for CCL4. To further verify these findings, we measured intracellular expression levels of these chemokines by FACS. CD8+ enriched T cells were incubated overnight with IL-12 and IL-18 and chemokine expression from the CD161++ Va7.2+ population was assessed (***Figure 4C***). Although only CCL4 was detectable by this method, the results showed a significant increase in CCL4 expression in IL-12/18 stimulated MAIT cells compared to the unstimulated controls (p = *0.0169*) confirming data from the stimulated MAIT supernatants (***Figure 4A and B***). Further, gating on all CCL4+ cells within the total CD8+ T cells, revealed that CD161++ Vα7.2+ cells are highly enriched within the CCL4-producing subset (***Figure 4D***), accounting on average for 60% of this population. Hence, MAIT cells likely represent the major source of CCL4 in our model system.

Subsequent profiling of the effector molecules produced by MAITs showed that they are also capable of producing CXCL12/SDF-1, the ligand for the alternative HIV-1 entry co-receptor, CXCR4. In contrast to CCL3, 4 and 5, CXCL12 induction required TCR signaling, as it was only observed in co-culture of MAIT cells with 5-OP-RU-loaded THP-1 cells (***Figure 4—figure supplement 1***). In accordance with these results, no anti-HIV-1 effect of IL-12/18-stimulated supernatants was observed when the CXCR4 tropic $HIV-1_{LAI}$ virus was used instead of $HIV_{BAL}$ to infect the target cells (***Figure 4—figure supplement 2***). Therefore, upon IL12/18 stimulation MAIT cells significantly increase expression of known HIV-1 restriction factors.

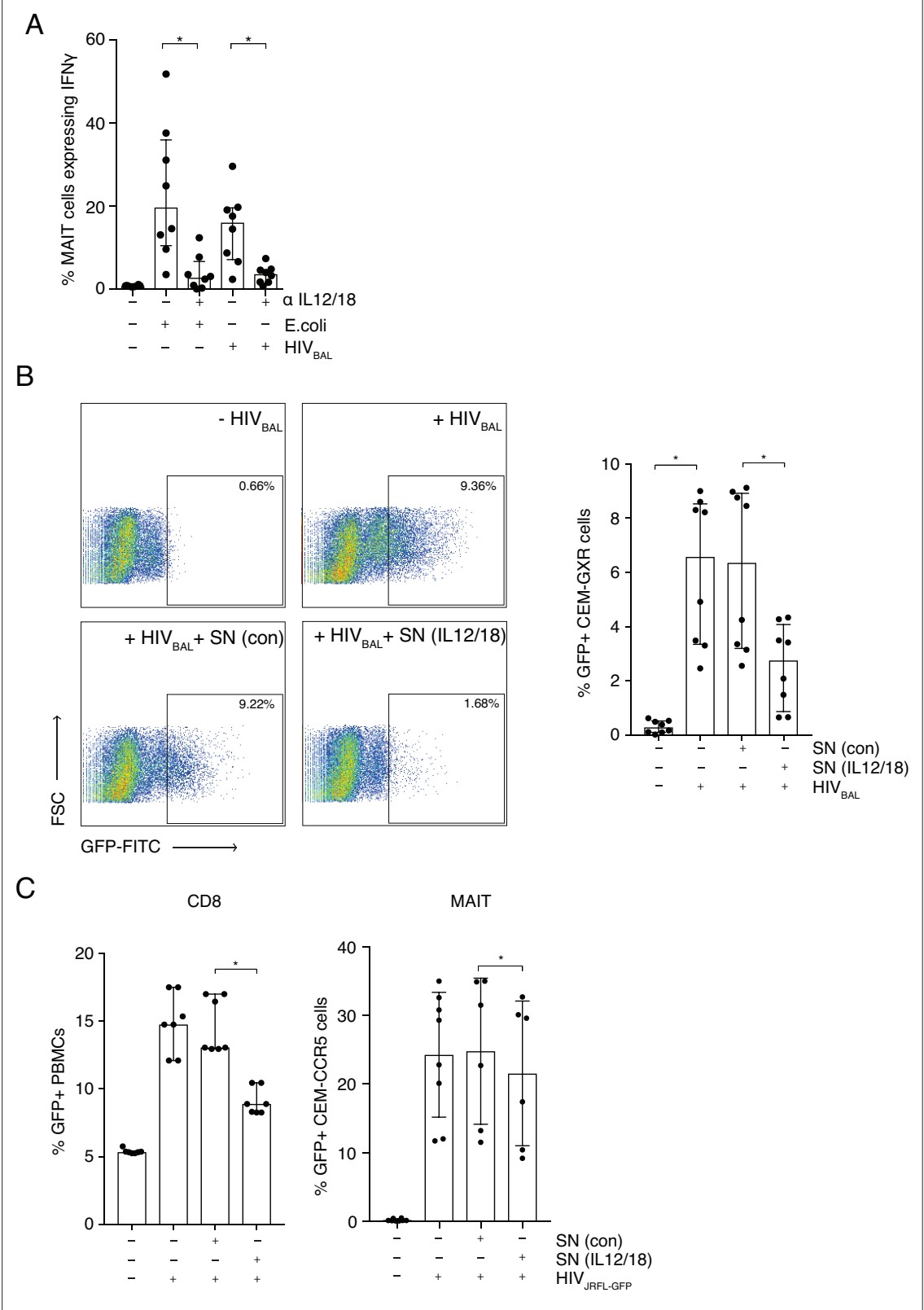

**Figure 3.** MAIT cells are activated by HIV-1 in an IL-12 and IL-18-dependent manner and display anti- HIV-1 activity. (**A**) Bar plots showing the percentage of MAIT cells expressing IFN-γ upon in vitro stimulation with fixed *E. coli* or HIV$_{BAL}$ in the presence or absence of blocking antibodies directed against IL-12 and IL-18. (**B**) Reduced frequency of GFP positive CEM-GXR cells following infection with HIV$_{BAL}$ (MOI = 0.2) and pre-treatment with stimulated supernatant from MAIT cells. Shown are representative dot plots (left) and cumulative column bars (right). (**C**) Inhibition of HIV$_{JRFL-GFP}$

*Figure 3 continued on next page*

*Figure 3 continued*

infection in primary human PBMCs or CEM-CCR5 cells by addition of control or IL-12/18-treated supernatants obtained from MACS-enriched CD8s (left) or FACS-sorted MAIT cells (right). *p < 0.05, paired t-tests. Data were pooled from three independent experiments; error bars indicate the standard deviation.

The online version of this article includes the following source data and figure supplement(s) for figure 3:

**Source data 1.** MAIT cells are activated by HIV-1 in an IL-12 and IL-18-dependent manner.

**Figure supplement 1.** Inactivated HIV-1 does not stimulate MAIT cells.

**Figure supplement 2.** Inhibition of HIV-1 is increased with higher volumes of stimulated supernatant.

**Figure supplement 3.** Increased expression of restriction factor and pro-apoptosis markers in MAIT cells stimulated with IL-12 and IL-18.

**Figure supplement 4.** MAIT cell anti-viral activity is not dependent on IFNγ.

**Figure supplement 5.** Inhibition of HIV-1 is not dependent on cell contact.

### Antiviral chemokines mediate the inhibition of HIV-1 infection by MAIT cells

As MAIT cells were able to produce three CCR5 binding factors, we aimed to determine their role in the anti-HIV-1 effect of MAIT cells. Using our CEM-GXR-GFP reporter cell - HIV$_{BAL}$ system, we added blocking antibodies to CCL3/4/5 and measured levels of viral infection. Strikingly, the addition of these blocking antibodies neutralized the inhibitory effect of the IL12/18-stimulated supernatant and restored GFP expression to levels equivalent to the control condition (*Figure 4E*). Taken together these data indicate that the anti-viral activity of MAIT cells in relation to HIV-1 is mediated at least in part by expression of the chemokines CCL3, CCL4, and CCL5 (*Figure 4—source data 1*).

## Discussion

The impact of HIV-1 on MAIT cell populations was initially noted by two groups in 2012, and depletion of MAIT cells in blood of HIV-1-infected individuals has been subsequently broadly confirmed with effects seen in acute, chronic and long-term treated infection, and reproduced here (*Eberhard et al., 2016*; *Cosgrove et al., 2013*; *Spaan et al., 2016*; *Paquin-Proulx et al., 2017*; *Leeansyah et al., 2013*). However, while such clinical studies have focused on the impact of HIV-1 on MAIT cells, none have described how MAIT cells can be activated by HIV-1 and most importantly a potential impact of MAIT cells on HIV-1 (*Ussher et al., 2018*). Given recent data indicating that viruses can activate MAIT cells efficiently in vivo (*van Wilgenburg et al., 2016*) and provide protection against lethal challenge in a mouse model (*van Wilgenburg et al., 2018*), we aimed to define whether MAIT cells can respond to and suppress HIV-1.

To address this, we first defined activation of MAIT cells in vivo. We observed strongly increased inhibitory receptor and granzyme B expression. Granzyme B-positive MAIT cells were detected during acute HIV-1 infection (PHI), up to 10-fold higher than HIV-1- controls. During chronic infection the majority of MAIT cells were granzyme B+, more than 90% for some individuals. These data are highly congruent with those seen during acute dengue and influenza infections (*Loh et al., 2016*; *van Wilgenburg et al., 2016*). This trend was similar when inhibitory receptors PD-1 and TIM-3 were examined. Higher levels of both inhibitory receptors were observed in CHI and to a lesser degree in PHI, and reduction of both receptors occurred following ART. This response to treatment is similar, although not identical to that seen in other virus infections. Residual MAIT cells in chronic HCV-infection showed an activated phenotype with high levels of granzyme B, HLA-DR, PD-1 and CD69 expression (*Hengst et al., 2016*) and granzyme B levels remain high after successful cure (*van Wilgenburg et al., 2016*). MAIT cells were not reinvigorated following HCV clearance and remained dysfunctional (*Hengst et al., 2016*). Sustained expression of granzyme B and PD-1 was also observed during tuberculosis (TB) and HIV-1/TB co-infection (*Jiang et al., 2014*; *Saeidi et al., 2015*). Blockade of PD-1 on MAIT cells during TB infection resulted in significantly higher IFNγ expression when stimulated with BCG. This is yet to be proven in HIV-1 infection and it remains to be seen whether immune checkpoint blockade can rescue MAIT cell function following chronic activation.

MAIT cells are known to migrate toward mucosal sites, recognise bacterial metabolites and contribute to barrier immunity (*Voillet et al., 2018*). One possible hypothesis for low MAIT cell

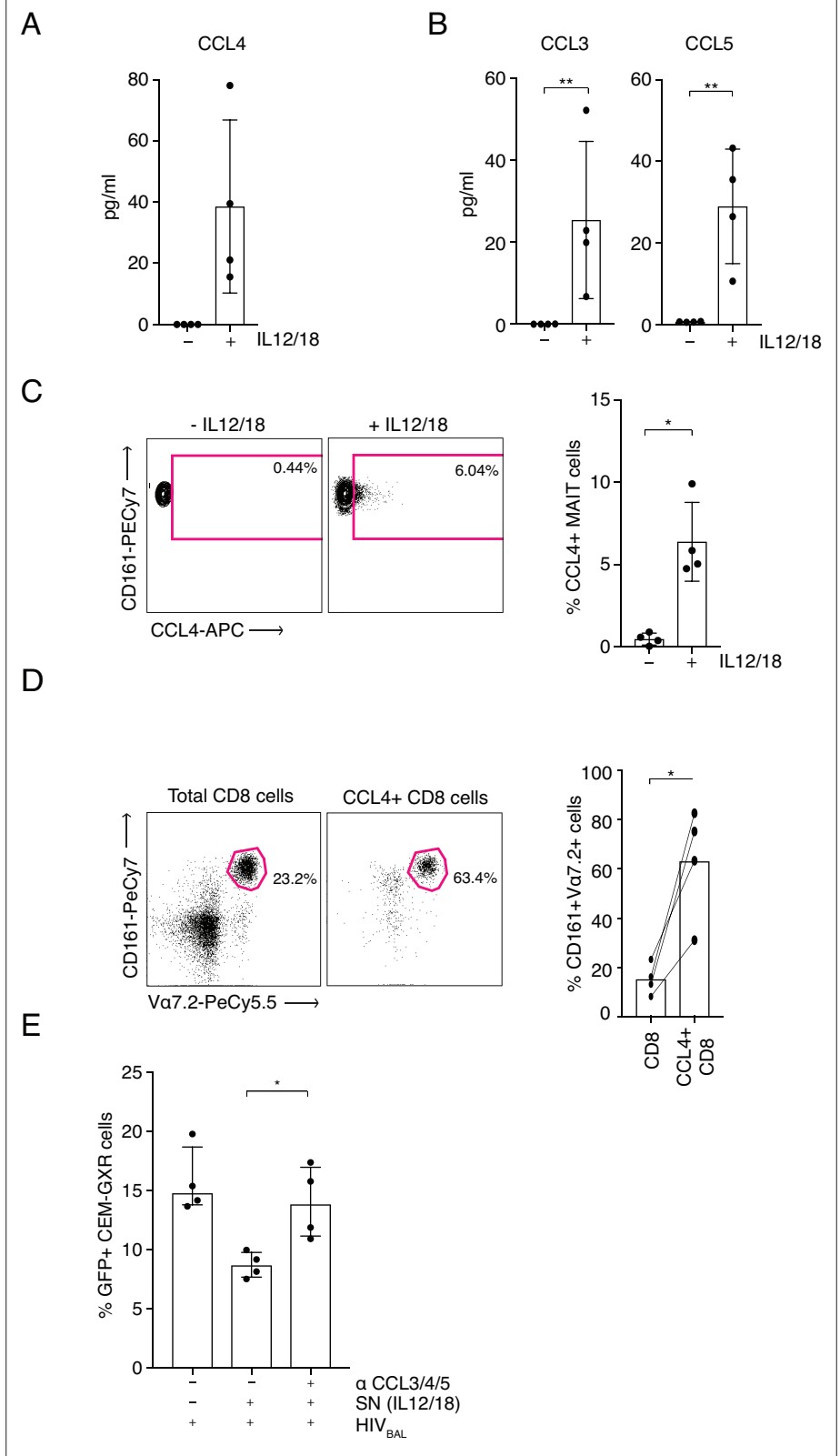

**Figure 4.** MAIT cell derived antiviral restriction factors are essential for suppressing HIV-1 in vitro. (**A**) MAIT cells were FACS-sorted and the CCL4 (MIP-1β) concentration was measured in the supernatants by ELISA after 20 hr post stimulation with IL-12/18. (**B**) MAIT cells were FACS-sorted and the concentrations of CCL3 (MIP1α) and CCL5 (RANTES) were measured in the supernatants by cytometric bead array (CBA) after 20 hr post stimulation

*Figure 4 continued on next page*

*Figure 4 continued*

with IL-12/18. (**C**) Representative FACS-plots (left) and bar plots (right) depicting the expression of CCL4 by MAITs after incubation with IL-12/18 for 20 hr. MAIT cells were identified as CD161++ Vα7.2+ cells within MACS-enriched CD8s. (**D**) Representative FACS dot plots (left) and bar plots (right) showing the percentage of MAIT cells as identified by co-expression of Vα7.2 with high levels of CD161 within MACS-enriched CD8s and within all CCL4-expressing CD8 T cells from the same culture. CD8 T cells were stimulated with IL-12/18 for 20 hr. (**E**) Recovery of GFP-positive CEM-GXR cells following blocking of restriction factors (CCL3/4/5), after treatment with IL12/18 stimulated supernatant from CD8 cells and infection with HIV$_{BAL}$. *p < 0.05, **p < 0.05, paired t-tests. Data were pooled from two independent experiments; error bars indicate the standard deviation.

The online version of this article includes the following source data and figure supplement(s) for figure 4:

**Source data 1.** MAIT-cell-derived antiviral restriction factors are essential for suppressing HIV-1.

**Figure supplement 1.** TCR-induced expression of CXCL12/SDF-1 by activated MAIT cells.

**Figure supplement 2.** MAIT cells do not inhibit infection by a CXCR4 tropic virus.

frequencies in peripheral blood is that they may migrate into mucosal tissues following infection and immune activation. MAIT cell frequencies in rectal tissue of treated PHI donors were not found to be substantially different from HIV-1- controls (*Figure 2F*). In this study, higher percentages were found within the rectum of HIV-1+ individuals, compared to terminal ileum. In pigtail macaques, MAIT cells exhibited lower expression of the gut homing marker α4β7 and were not enriched in the gut prior to SIV infection. Following infection, they upregulated α4β7 and their frequencies increased within the rectum (*Juno et al., 2019b*). The authors postulated that this may prevent depletion of MAIT cells during chronic infection. Here, a positive correlation was observed when blood and tissue MAIT cells were compared, although MAIT cell percentages were lower overall in blood. This suggests that there is a general reduction of MAIT cells in both compartments. If the impact of HIV-1 on MAIT cells was primarily redistribution to gut, then an inverse relationship between frequencies at the two sites would be expected.

To elucidate whether HIV-1 can activate MAIT cells, an R5 tropic lab strain virus (HIV$_{BAL}$) was used to stimulate THP1 monocytes co-cultured with CD8+ T cells. IFN-γ expression in MAIT cells was observed when HIV-1$_{BAL}$ was used to stimulate THP-1 antigen presenting cells but had no anti-HIV-1 effect. The THP-1 monocytic cell line was used as it has been extensively trialled by our group and many others as an effective APC for MAIT cells in microbial and viral infections (*Ussher et al., 2018*; *van Wilgenburg et al., 2018*). MAIT cell activation was dependent on IL-12 and IL-18 produced by THP1 cells via innate sensing (*van Wilgenburg et al., 2016*; ). THP1 sensing of HIV-1$_{BAL}$ may be through TLR7/8 or cytosolic RIG-like receptors (*Diget et al., 2013*; *Guo et al., 2014*), which in turn activate THP-1 cells to secrete inflammatory cytokines such as IL-12 and IL-18 (*Bandera et al., 2018*). Inactivated virus was incapable of activating THP-1 cells. This could be due to the cross-linking of nucleocapsid P7 protein by aldrithiol-2, which may not allow single stranded HIV-1 RNA within virions to bind to TLR7/8 (*Rossio et al., 1998*). We note that inactivation of influenza and HCV also impacted on MAIT cell recognition in vitro (*van Wilgenburg et al., 2018*), even in a macrophage culture where true infection and replication does not occur.

IFN-γ production dependent on IL-12 and IL-18 has also been observed in other infection models (*Okamura et al., 1998*), as well as dengue virus infection (*Fagundes et al., 2011*). It must be noted that stimulation of MAIT cells by IL-12 and IL-18 not only activates the cells independent of TCR signalling to produce pro-inflammatory cytokines and cytotoxic granules, but it also increases caspase-3 expression, a pro-apoptotic marker. It has been shown that IL-12 can induce apoptosis of CD8+ T cells in the absence of antigenic stimulation (*Fan et al., 2002*) and it has also been observed that MAIT cells show PLZF-dependent enhanced capacity for apoptotic death following stimulation (*Gérart et al., 2013*). Overall HIV-1-induced depletion of MAITs could very likely be due to activation induced apoptosis, as observed in other disease settings (*Wakao et al., 2017*; *Hinks, 2016*).

HIV-1 restriction factors CCL3 (MIP-1α), CCL4 (MIP-1β), and CCL5 (RANTES) were detected in supernatants of purified MAIT cells stimulated in vitro. This fits with recent data exploring the broad activity of MAIT cells following diverse stimuli in mice and humans (*Hinks et al., 2019*; *Leng et al., 2019*; *Leng et al., 2019*). Such chemokines may play a role in cellular chemoattraction following infection consistent with the early recruitment role for MAIT cells observed in infectious models (*Meierovics et al., 2013*; *van Wilgenburg et al., 2018*). Stimulated supernatants containing these

restriction factors were able to inhibit HIV-1 infection of target cells in a specific manner, and the inhibition observed was not dependent on cell contact. Loss of viral inhibition was observed when stimulated supernatants were treated with restriction factor-blocking antibodies before addition of target cells. Collectively, this data suggests that MAIT cells can be activated by HIV-1 via a TCR-independent pathway. IL-12 and IL-18 are necessary for MAIT cell activation and secretion of restriction factors in vitro, and activation induced cell death may explain declining numbers of MAIT cells during HIV-1 infection.

Both MAIT cells and THP-1 cells can produce a broad range of factors following stimulation as has been recently described (*Hinks et al., 2019*; *Lamichhane et al., 2020*; *Leng et al., 2019*). However, even though factors such as IFNγ could potentially possess anti-HIV-1 activity, as reported for iNKT cells (*Paquin-Proulx et al., 2016*), we were able to block the suppressive activity very effectively with anti-chemokine antibodies, suggesting that at least in our models, these are the most potent effectors. There has been no significant difference in HIV-1 specific IFNγ response reported in both progressors and long-term non-progressor patients with chronic HIV-1 infection (*Roff et al., 2014*; *Zanussi et al., 1996*). This is consistent with original in vitro studies which revealed no antiviral effect of IFNγ and even enhancement of infection in primary cells (*Yamamoto et al., 1986*; *Mackewicz et al., 1994*) and several clinical trials that revealed no impact of this cytokine in vivo (*Roff et al., 2014*).

These data provide novel evidence that MAIT cells can respond to HIV-1 and that part of this activation includes an antiviral function. It is not known whether activation of MAIT cells in a TCR-independent manner can lead to cell killing as this will likely depend on the nature of the interaction, including a potential role for cell surface interactions analogous to those involved in NK cell activation. In the GI tract it is also possible that high local levels of the MAIT cell ligand (5 OP-RU) could pre-sensitise MAIT cells via the TCR to enhance functionality and responsiveness to cytokines, increasing the levels of antiviral chemokines. Even in the absence of killing it is clearly evident that CCR5 levels play an important role in the long-term outcome of HIV-1 infection (*McLaren et al., 2015*) and thus this effect could certainly contribute to control of HIV-1 levels in vivo. More intriguingly, since MAIT cells are present in relevant tissues in persistent infection, including under therapy, it will be important to consider their role in strategies for cure.

Overall, these data, taken with other findings, suggest that the strong activation of MAIT cells by HIV-1 in natural infection can be accompanied by relevant antiviral functions. The recent data showing that MAIT cells can protect against lethal viral infection in mice (where they exist in much lower frequencies) provides some proof-of-principle that such activity can play an important role in vivo, in conjunction with other cell types. In the absence of a small animal model for HIV-1 where MAIT cells can be depleted, this sort of proof will be hard to reproduce for this infection, but further evidence should be sought to support this hypothesis.

# Materials and methods

## Participant samples

Participants with Primary HIV-1 (PHI) were recruited as either part of the HEATHER (HIV-1 Reservoir targeting with Early Antiretroviral Therapy) cohort or from the SPARTAC (Short Pulse Antiretroviral Therapy at HIV-1 Seroconversion) trial. For inclusion in the HEATHER cohort, participants with identified PHI commenced ART within 3 months of diagnosis, and did not have co-infection with Hepatitis B or C. For our study, cryopreserved PBMCs were used from the closest pre-therapy sample to seroconversion (baseline) and from a sample 9–15 months after commencement of ART (1 year). Only Baseline samples were used from the SPARTAC trial, which was a multi-centre, randomised controlled trial of short course ART during PHI, the full design of which is described elsewhere (*Fidler et al., 2013*).

Participants with Chronic HIV-1 (CHI) were recruited in Bloemfontein, located within the Mangaung Metropolitan Municipality in the Free State province of South Africa. Most participants had advanced HIV-1 disease progression (as reflected by a CD4 T cell count <350 cells/μL). All participants were tested for HIV-1 using a point-of-care 'HIV-1 rapid test' or laboratory-based HIV-1 ELISA. Follow-up samples were collected at 6- and 12 months post-ART initiation.

Long-term non-progressor (LTNP) samples were collected at various sites across New South Wales, Australia; samples were processed and stored at St. Vincent's Centre of Applied Medical Research, Darlinghurst. Eligible subjects were HIV-1+, asymptomatic, diagnosed at least 8 years prior to

**Table 1.** Participant cohort characteristics.
CD4 T cell count and HIV-1 (log) viral load.

| Patient cohort | Sample Number | CD4+ T cells (Count/µL) median (IQR) | Plasma viral load (log10 copies/mL) median (IQR) |
|---|---|---|---|
| Healthy Donors | 12 | - | - |
| PHI (SPARTAC) | 8 | 596 (437–755) | 5.04 (4.51–5.45) |
| PHI (HEATHER) | 12 | 524 (437–656) | 4.34 (3.19–4.88) |
| HEATHER GUT | 11 | - | < 1.3 |
| CHI | 12 | 360 (72–646) | 4.90 (3.55–5.59) |
| EC LTNP | 9 | 780 (615–1013) | 1.70 (1.60–1.70) |
| VC LTNP | 10 | 633 (442–800) | 5.18 (5.03–5.37) |

enrolment, treatment naïve, and had an absolute CD4+ T cell count ≥500 cells/µL. Elite controllers (EC) had an undetectable viral load (median <1.7 Log) whilst viraemic controllers (VC) had a detectable viral load (median <5.8 Log) (*Table 1*).

## Processing of tissue samples

Rectal and terminal ileum biopsies (up to 12 from each site) were collected at endoscopy and immediately placed in complete media [RPMI-1640 media with 5% heat-inactivated fetal bovine serum (FBS), 0.04 mg/mL gentamicin, 100 IU/mL penicillin, 0.1 mg/mL streptomycin and 2 mM L-glutamine] and processed within 3 hours of sampling. Briefly, samples were washed in 1 mM dithiothreitol (DTT) solution and then with PGA solution (Hanks' Balanced Salt Solution with 0.04 mg/mL gentamicin, 100 IU/mL penicillin and 0.1 mg/mL streptomycin). Biopsy samples subsequently underwent collagenase and mechanical digestion using Collagenase D (1 mg/mL) for 30 min and a gentle MACS dissociator (Miltenyi Biotec), respectively. The resulting cell suspension was then strained using a 70 µM filter, washed with a penicillin/streptomycin, glutamine and amphotericin (PGA) solution containing 500 ml Hank's Balanced Salt Solution (HBSS) without $Ca^{2+}$ and $Mg^{2+}$, 5 ml penicillin (100 IU/mL) /streptomycin (0.1 mg/mL), 2 ml Gentamicin, (10 mg/mL) Amphotericin B (50 uL). The washed cells were then used for staining.

## Flow cytometry

Frozen PBMC were thawed using R10 medium (RPMI + L-glutamine+ Penicillin Streptomycin + 10% FCS) and subsequently stained with antibodies corresponding to either the chemokine/cytokine receptor, cytotoxic, or transcription factor panels (see below). A FoxP3 permeabilization kit (BD Pharmingen) was used for intracellular/intranuclear staining. Staining of the chemokine panel was carried out at 37 °C. Samples were acquired on an LSRII flow cytometer (BD Biosciences) using the FACSDiva software package (BD Biosciences). Prior to each run, all samples were fixed in 2% PFA. Samples were then analyzed using the Flowjo software package (FlowJo, LLC). Gating strategies were developed based on florescence-minus-one (FMO) controls.

### Base panel

Live/Dead dye (Invitrogen), CD4 (RPA-T4, BD Biosciences), CD3 (UCHT1), CD8 (5K1), Vα7.2 (3C10), and CD161 (191B8) [all Biolegend].

### Activation/inhibitory receptor panel

GzmB (GB11), PD-1 (EH12.1) and TIM-3(7D3) all Biolegend.

## CD8 MACS enrichment

CD8 T cells where enriched from whole PBMCs using a CD8 microbead-positive selection kit (Miltenyi) according to the manufacturer's instructions. Briefly, whole PBMCs were isolated, washed in MACS buffer and incubated with CD8 microbeads for 15 min on ice. Cells were washed again and run over

magnetic LS selection columns placed in a MACS magnet. After three rinsing steps, the columns were removed from the magnet and the enriched CD8 T cells were eluted from the columns. Enriched CD8 T cells were washed, counted and subjected to downstream experiments. A fraction of the enriched CD8 T cells were stained with Live/dead dye and CD8 antibodies to determine purity by FACS staining. On average enriched CD8 T cells had a viability >95% and a purity >96%.

## MAIT cell sorting

To obtain pure MAIT cells for functional analyses, MACS-enriched CD8s were isolated and stained for FACS-sorting with the following antibodies and dyes: CCR6 (Biolegend, G034E3), CD161 (Miltentyi, 191B8), CD8 (Biolegend, SK1) and near-infrared Live/Dead dye (Invitrogen). To avoid TCR-triggering by antibodies, no antibodies against Vα7.2 and CD3 were included in the sorting panel and MAIT cells were instead identified by surrogate markers as CD161$^{hi}$ CCR6$^+$ cells. Sorting was performed on a MA-900 sorter (SONY) using a 100 μm sorting chip. To validate the sorting strategy and to determine the purity of the cells a small fraction of each sorted sample was stained with antibodies against CD3 and Vα7.2 as well as Live/Dead dye. Viability of sorted MAIT cells was on average >93% and > 98% of the sorted cells stained positive for Vα7.2. Samples, which contained considerable populations of dead ( > 15%) or Va7.2 negative ( > 10%) cells were excluded from downstream experiments.

## Cell lines

THP1 (ATCC), CEM-GXR (NIH AIDS Reagent Program), CEM-RR5 and Jurkat-Tat-R5 (both kindly provided by Quentin Sattentau) were maintained in RPMI 1640 media containing 10% FCS, L-glutamine and penicillin/streptomycin (Sigma-Aldrich). Cells were cultured every 3–4 days and incubated at 37 °C and 5% CO$_2$. All cell lines used were mycoplasma negative.

## In vitro stimulations of MAIT cells

For the TCR-specific triggering of MAIT cells, PBMC-derived CD8 T cells were enriched over MACS columns and $2 \times 10^5$ cells were co-cultured with $1 \times 10^5$ THP1 (ATCC) cells which had been pulsed with 10 nM 5-OP-RU for 2 hr. Unpulsed THP1s were used as controls. CD8s and THP1s were cultured together for three or 6 days and the expression of CXCL12 in MAIT cells was assessed by flow cytometry. To ensure that the observed effects were TCR-mediated, an anti-MR1 blocking antibody (5 μg/mL, clone 26.5, Biolegend) was added to control wells. For cytokine-induced MAIT cell activation, $2 \times 10^5$ MACS-enriched CD8s or FACS-sorted MAIT cells were stimulated overnight with IL-12 and IL-18 (both at 50 ng/mL). The supernatants of these cultures were harvested and stored at –80 °C before being used to treat HIV-1-BAL infected cells or to assess the expression of viral restriction factors. Supernatant from unstimulated cultures was used as controls.

## Intracellular cytokine staining assay

$2 \times 10^5$ MACS-enriched CD8 cells were incubated overnight with or without IL-12 and IL –18 (both at 50 ng/mL) at 37 °C. For the last 4 hr of incubation, Brefeldin A and monensin (Biolegend) were added to block the release of cytokines. Cells were harvested and permeabilized with Cytofix/perm (Becton Dickinson) and stained with intracellular antibodies to base panel as described above, and IFN$\gamma$ (B27, Biolegend), CCL4 (MIP-ß, FL3423L. BD or REA511, Miltentyi), Caspase-3 (C92-605, BD), CXCL12/SDF-1 (79018, R&Dsystems). MAIT cells were identified as CD161$^{HIV-1}$α7.2+ cells.

## Cytometric bead array and ELISA

A total of $2 \times 10^5$ FACS-sorted MAIT cells were incubated overnight with or without IL-12 and IL –18 (both at 50 ng/mL) at 37 °C. Cells were pelleted by centrifugation and the supernatants of the cultures was harvested to assess MAIT-specific production of CCL3,4 and 5. CCL3 and 5 production was analyzed by a Cytometric bead array (BD) following the manufacturer's instructions. Briefly, 50 μl of undiluted supernatant was iteratively mixed and incubated with beads and detection reagents specific for CCL3 or CCL5 and the resulting fluorescent signals were recorded on a LSR II flow cytometer (BD). Protein concentrations were calculated from these fluorescent values based on standard curves generated in each experiment from serially diluted CCL3 and CCL5-stocks with defined concentrations. The concentration of CCL4 in each supernatant was determined using a DuoSet ELISA kit (R&D) following the manufacturer's instructions. The supernatant was serially diluted and a one in two dilution was

used for the final analysis. The concentration was determined from a standard curve which was run in parallel.

## In vitro virus experiments

The THP1 monocytic cell line was incubated with the HIV-1 lab strain $HIV_{BAL}$ or aldrithiol (AT-2) inactivated $HIV_{JRFL}$ virus at an MOI of 0.2 for 6 hr at 37 °C. Positively bead (Thermo Scientific) selected CD8+ T cells were then added to the culture overnight at a ratio of 1:1 with THP1 cells. Anti-IL-12 and anti-IL-18 blocking antibodies (R&D) were used at 10 µg/mL and added to the culture prior to the addition of CD8+ cells. Formaldehyde-treated TOP10 *E. coli* cells (Thermofisher) were added to THP1 cells as a positive control. IFNɣ (B27, Biolegend) expression was then measured on an LSRII cytometer (BD biosciences).

HIV-1 inhibition assay- CEM-GXR cells, which express GFP under the control of the HIV-1 promoter, CEM-CCR5 cells or Jurkat-Tat-R5 cells were used for the inhibition assays as they express endogenous levels of CXCR4 and have been engineered to overexpress CCR5. Cells were incubated with unstimulated, or IL-12/18 stimulated supernatant for 6 hr at 37 °C. $HIV-1_{BAL}$, $HIV_{JRFL-GFP}$ or $HIV_{LAI}$ was added to the culture at an MOI of 0.2. Where indicated, anti-CCL3/4/5 were added at a concentration of 10 µg/mL and anti-IFNγ (clone B27, Biolegend) at a concentration of 50 µg/mL. Expression of GFP or p24 (KC57, Beckman Coulter) was measured 4 days post infection by FACS.

Healthy donor PBMCs were incubated with unstimulated or IL-12/18-stimulated supernatant for 6 hr at 37 °C and $HIV_{JRFL-GFP}$ primary viral isolate was added to the culture. GFP expression was analyzed at day 7 on a BD LSRII.

Transwell - For cell contact experiments, MAIT cells were either co-cultured with CEM-GXR cells or separated using Transwell plates (Corning). $HIV_{BAL}$ was then added and GFP was measured at day 4.

## Statistical analysis

Ex vivo data from six or more healthy donors and HIV-1+ donors (PBMC and Tissue), and four or more from in vitro stimulated samples were used for statistical calculations. All column graphs are presented as medians with inter-quartile ranges. Wilcoxon paired *t* test was used to analyze statistical data employing Prism 7.0 (GraphicPad, La Jolla, CA, USA) software. For unpaired samples the Mann-Whitney U test was used. p-Values < 0.05 were considered significant (* < 0.05, ** < 0.01, *** < 0.001, and **** < 0.0001).

## Acknowledgements

We thank Dr Vera Klemm for proofreading our manuscript. PK and CP are supported by the Wellcome Trust (WT109965MA), NIHR Senior Fellowship (PK), and the NIHR Biomedical Research Centre, Oxford. JM and DF are supported by the ARC (CE140100011) and NHMRC (1117017). We thank all healthy donors and LTNP participants. We thank Jeff Lifson for the AT-2 inactivated virus stocks. We thank the participants of SPARTAC and HEATHER. The HEATHER study is conducted as part of the CHERUB (Collaborative HIV-1 Eradication of Reservoirs): (UK BRC) collaboration. (CHERUB Steering Committee): Andrew Lever (University of Cambridge), Mark Wills (University of Cambridge), Jonathan Weber (Imperial College, London), Sarah Fidler (Imperial College, London), John Frater (University of Oxford), Lucy Dorrell (University of Oxford), Mike Malim (King's College, London), Julie Fox (King's College London), Ravi Gupta (University College London), Clare Jolly (University College London).

## Additional information

### Funding

| Funder | Grant reference number | Author |
| --- | --- | --- |
| Wellcome Trust | WT109965MA | Paul Klenerman |

The funders had no role in study design, data collection and interpretation, or the decision to submit the work for publication.

## Author contributions
Chansavath Phetsouphanh, Conceptualization, Data curation, Funding acquisition, Investigation, Methodology, Project administration, Validation, Writing – original draft, Writing – review and editing; Prabhjeet Phalora, Carl-Philipp Hackstein, Data curation, Investigation, Methodology, Writing – review and editing; John Thornhill, C Mee Ling Munier, Jodi Meyerowitz, Lyle Murray, Cloete VanVuuren, Dominique Goedhals, Rebecca A Russell, Quentin J Sattentau, Jeffrey YW Mak, Sarah Fidler, Methodology, Resources; Linnea Drexhage, Methodology; David P Fairlie, Anthony D Kelleher, Methodology, Resources, Writing – review and editing; John Frater, Conceptualization, Supervision; Paul Klenerman, Conceptualization, Resources, Supervision, Writing – original draft, Writing – review and editing

## Author ORCIDs
Chansavath Phetsouphanh http://orcid.org/0000-0001-6617-5995
Cloete VanVuuren http://orcid.org/0000-0002-9095-0039
Quentin J Sattentau http://orcid.org/0000-0001-7170-1937
John Frater http://orcid.org/0000-0001-7163-7277
Paul Klenerman http://orcid.org/0000-0003-4307-9161

## Ethics
Clinical trial registration 2004-000446-20.
Human subjects: The SPARTAC trial (EudraCT Number: 2004-000446-20) was approved by the following authorities: the Medicines and Healthcare products Regulatory Agency (UK), the Ministry of Health (Brazil), the Irish Medicines Board (Ireland), the Medicines Control Council (South Africa) and the Uganda National Council for Science and Technology (Uganda). It was also approved by the following ethics committees in the participating countries: the Central London Research Ethics Committee (UK), Hospital Universitário Clementino Fraga Filho Ethics in Research Committee (Brazil), the Clinical Research and Ethics Committee of Hospital Clinic in the province of Barcelona (Spain), the Adelaide and Meath Hospital Research Ethics Committee (Ireland), the University of Witwatersrand Human Research Ethics Committee, the University of Kwazulu-Natal Research Ethics Committee and the University of Cape Town Research Ethics Committee (South Africa), Uganda Virus Research Institute Science and ethics committee (Uganda), the Prince Charles Hospital Human Research Ethics Committee and St Vincent's Hospital Human Research Ethics Committee (Australia) and the National Institute for Infectious Diseases Lazzaro Spallanzani, Institute Hospital and the Medical Research Ethics Committee, and the ethical committee of the Central Foundation of San Raffaele, MonteTabor (Italy). Recruitment for and studies within the HEATHER cohort were approved by the West Midlands- South Birmingham Research Ethics Committee (reference 14/WM/1104). Recruitment of CHI participants were approved by The University of the Free State Ethics Committee (ETOVS 171/08). LTNPs recruitment was approved by The St Vincent's Human Research Ethics Committee (EC00140) approval number: HREC/12/SVH/298, SVH 12/217. PBMCs obtained from healthy donors were approved by the Sheffield Research Ethics Committee (reference 16/YH/0247). Participants were aged 18 years or older, all participants from each of the above mentioned cohorts gave informed and written consent for their participation in these studies.

## Decision letter and Author response
Decision letter https://doi.org/10.7554/eLife.50324.sa1
Author response https://doi.org/10.7554/eLife.50324.sa2

# Additional files

## Supplementary files
• Transparent reporting form

## Data availability
Raw data from main figures are provided as source data files.

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
