## [Decision Letter]

**Decision letter after peer review:**

Thank you for sending your article entitled "Human MAIT cells respond to and suppress HIV" for peer review at *eLife*. Your article is being evaluated by 3 peer reviewers, one of whom is a member of our Board of Reviewing Editors, and the evaluation is being overseen by Tadatsugu Taniguchi as the Senior Editor.

Summary:

The study by Phetsouphanh et al. deals with the role of MAIT cells in HIV infection. As shown previously, the authors found that MAIT cells are activated/exhausted in primary and chronic HIV infection, but, at the same time, their numbers are dramatically reduced. in vitro stimulation of MAIT cells with HIV infected THP-1 monocytes was used to characterize the mechanism of MAIT cell activation. Infected cells secrete IL-12 and IL-18 which then stimulate MAIT cells for production of effector molecules which inhibit HIV. Namely, MAIT cell secretion of chemokines is sufficient to inhibit HIV infection of a CCR5-overexpressing CEM cell line. In general, the study is relevant and interesting. However, too many open issues need serious additional work. Having in mind *eLife*'s policy not to accept the papers that require substantial interventions, we need the authors to let us know a realistic expectation that requested experiments will be completed in 2 months. If the authors find unrealistic that they can revise the paper within two months, we advise the authors to resubmit the manuscript after finishing all the requested experiments. Please note that in that case the resubmitted manuscript will be considered as a new submission.

Essential revisions:

1. It is well established through multiple studies that MAIT cells (defined by surface marker expression) are depleted during HIV infection and poorly reconstituted during ART. Additionally, changes in GzmB and Tim-3 expression, as well as general MAIT cell activation, during HIV infection have previously been reported (Leeansyah 2013, Leeansyah 2015, and others). It is surprising that the current study does not take advantage of MR1 tetramers to identify the MAIT cell population, as this would present the opportunity to characterize the MAIT population in HIV-infected cohorts in a manner which would be more novel. The data on MAIT cells in the GALT would be stronger if representative staining were shown, but would also benefit immensely from use of the MR1 tetramer to characterize GALT MAIT cells, as this has not previously been done.

2. Infection of THP-1 monocytes in vitro was used to show that HIV can stimulate MAIT cells via IL-12 and IL-18. The use of THP-1 as the target cells for HIV-1 infections was somewhat intriguing since THP-1 cells usually don't support significant productive HIV-1 infection. With this in mind, it would be better to use cells that support productive HIV-1 BaL infection eg. monocyte-derived macrophages (MDM). It remained unclear are IL-12 and IL-18 produced by infected or bystander THP-1 monocytes? Can inactivated HIV induce IL-12 and IL-18? IL-12/18 produced by THP-1 cells themselves might also have some antiviral effects through various MAIT cell-independent mechanism. Have the authors ruled out this potential effect? Are the levels of IL-12 and IL-18 elevated in HIV+ patients in blood and in tissues?

3. The data shown in Figure 3 does clearly demonstrate that exposure of THP-1 cells to HIV results in IL-12/IL-18-driven MAIT cell activation in vitro. It is important to note, however, that this assay recapitulates only a very specific aspect of HIV infection, and does not include any assessment of MAIT cell activation by HIV-infected CD4^+^ T cells or other mechanisms such as microbial translocation. It is surprising that MAIT cell production of IFNγ was not attributed any anti-viral role, considering that similar experiments with iNKT cells have shown that iNKT-derived IFNγ limits HIV infection of primary T cells in vitro (Vasan 2007). Considering recently published data indicating the CCL3 and CCL4 production by MAIT cells is significantly greater following TCR-mediated stimulation than cytokine stimulation (Lamichhane 2019), and the impact that microbial translocation could have on MAIT cell activation, it is unclear what the relative contribution of cytokine-based activation would be on MAIT cells in vivo during HIV infection.

4. Did the authors also see contact-dependent MAIT cell-mediated inhibition of HIV replication? Similarly, would TCR-activated MAIT cells mediate antiviral activity through CCL3-5 production and/or contact-dependent mechanisms?

5. Technically, there is no indication of the use of uninfected THP-1 cultures as a control in Figure 3A or B, and it is unclear as to whether the data presented in Figure 3B is background subtracted. Sorting on CD8^+^ T cells prior to culture with the THP-1 cells indicates that only CD8^+^ MAIT cells were studied in this experiment, which is a caveat that should be noted, as MAIT cells can include a substantial CD4-CD8- population as well.

6. The assay in Figure 4 using the CEM-GXR cells is likely biased to detect an impact of β-chemokines on HIV entry, given the cell line overexpresses CCR5. However, it would be important to confirm MAIT cell-mediated anti-HIV activity using primary human immune cells that productively support HIV-1 BaL replication, such as MDM and/or PHA/IL-2-activated PBMCs, plus measurements of productive viral replication (eg. p24, RT activity, etc).

7. Have the authors tested other R5 strains apart from BaL? It would be important to see that this antiviral effect is not just restricted to the BaL strain of R5 HIV-1. Along this line, there were only n=4 of presumably independent MAIT cell donors in Figure 4B (or was it 4 different CEM-GXR expts but using a single or pooled MAIT cell donors?). It would be good to increase the sample size to strengthen the manuscript. Similarly, have the authors tested X4 strains, lab-adapted or primary isolates? It would be an important control and interesting to see if MAIT cells can also/cannot inhibit X4 viruses. Along this line, did MAIT cells produce CXCR4-binding chemokines (eg. CXCL12/SDF-1)?

8. The authors show that IL-12 + IL-18 stimulation induces apoptosis in MAIT cells. While this is interesting observation, it does not explain why there is no increase in MAIT cell numbers following years of ART treatment. One can hypothesize that due to lower HIV load there will be less of IL-12 and IL-18, and subsequently less of MAIT cell stimulation. This is the case with the expression of GzmB, PD-1 and Tim-3. However, the numbers of MAIT cells remain the same. This discrepancy should be commented.

9. The methods section lacks important details for several experiments. The *E. coli* stimulations do not appear to be described at all. There is no indication of the number or ratio of T cells used in the THP-1 co-culture experiments.

[Editors' note: further revisions were suggested prior to acceptance, as described below.]

Thank you for submitting your article "Human MAIT cells respond to and suppress HIV" for consideration by *eLife*. Your article has been reviewed by 2 peer reviewers, and the evaluation has been overseen by a Reviewing Editor and Tadatsugu Taniguchi as the Senior Editor. The reviewers have opted to remain anonymous.

Although several of the requested data have been added, thus improving the quality of the manuscript, the reviewers pointed out that major concerns raised were not fully addressed to warrant sufficient novelty for publication in *eLife*. Additionally, on several occasions in response to requests from reviewers the authors stated what they could do, instead of dealing with the specific requests raised by reviewers. Since we still believe that your data showing that MAIT cells can inhibit HIV infection via secretion of R5 chemokines is worth considering for *eLife*, we decided to offer you another chance to fully deal with all reviewer's concerns, providing that you can assure us that all requested experiments will be conducted and re-revised manuscript submitted in frame of 2 months.

Below please find the summary of major concerns raised by reviewers.

1) The data in Figures 1 and 2 remain to be only confirmatory of multiple previously published studies. Without the data on GALT cells studied with MR1 tetramer, this portion of the manuscript does not add new information to the field. The authors argument that "this study commenced prior to these tetramers becoming available for staining, so for consistency we pursued this throughout" is somewhat in contradiction with their sentence stating that they have recently published the paper using MR1 tetramers to demonstrate MAIT cells in a GI tract. Some attempt at providing a confirmation of data should be done.

2) The response to reviewer's suggestion to assess the cells that support productive HIV-1 infection to stimulate MAIT cells instead of THP-1 cells, was also not clear. The authors state: "We can clarify the background to this experiment further" or "We can however readily test further whether combinations of monocyte-derived macrophages and BAL also work to activate MAIT cells in vitro, as previously tested for the other viruses …". Still, it remained unclear whether the authors performed the requested experiment and if so, it should be included in the manuscript. Since inactivated HIV did not result in THP-1 activation of MAIT cells, the question is what mechanism is behind this finding. This was not further discussed in the manuscript.

3) Potential anti-viral role of IFNγ was not explored. The authors state: "Regarding the IFNγ, we can certainly test this using blockade as we have done in other settings (eg HCV in vitro and influenza in vivo)."; Again, it is unclear whether or not these experiments have been performed. Although we agree that it is unlikely that IFNγ would result in a block of viral entry, it would still be relevant to test the impact of other MAIT cell-secreted cytokines and chemokines on viral replication, as the presence of MAIT cells at mucosal surfaces is clearly insufficient to block HIV infection. If the authors believe that mechanisms are the same in case of other viruses, we do not see the point of this study.

4) Control experiments using X4 viruses will further strengthen the manuscript, given the inhibition by MAIT cells is acting on R5 viruses (and thus no inhibition should be seen on X4 viruses).

[Editors' note: further revisions were suggested prior to acceptance, as described below.]

Thank you for submitting your revised manuscript '"Human MAIT cells respond to and suppress HIV' to *eLife*. Your manuscript was evaluated by three reviewers and the evaluation has been overseen by a Reviewing Editor and a Senior Editor. Although the two original reviewers were satisfied with the new results built into the revised manuscript, a third reviewer pointed to several weaknesses of the study.

The main objective of this reviewer is that the data provided so far do not fully support the conclusions made by the authors. Specifically, the reviewer emphasized the experiments involving stimulation of MAIT cells – under the experimental setting described only a portion of these cells belong to MAIT, while the vast majority are classical cytotoxic T lymphocytes. According to the reviewer, this fact calls into question the conclusions about the specific role of MAIT cells in HIV infection. The reviewer, therefore, requests the experiment on sorted MAIT cells instead of total CD8 T cells to exclude the possibility that CCL3/4/5 chemokines are produced by non-MAIT CD8 cells in response to IL-12/18.

Since during the consultations between the reviewers a unanimous opinion was not reached, the senior editor and the reviewing editor concluded that the above-specified request by the third reviewer was not entirely unfounded. In other words, although we are in favor of the final acceptance of the manuscript for publication in *eLife*, we request the authors to provide evidence that stimulated IL-12/18 sorted MAIT cells secrete CCL3/4/5 chemokines and demonstrate their function in functional assays.

Apart from above, one of the reviewers requested following corrections: Figure 3—figure supplement 3 – The units for CCL3-5 are in pg/mL whereas for CXCL12 is in gMFI. The authors should paraphrase the related sentence as these two units cannot be directly compared to each other. Otherwise, show the CXCL12 unit in pg/mL also.

*Reviewer #2:*

The authors have addressed my concerns.

*Reviewer #3:*

The authors have responded to the previous reviewer concerns and substantially improved the manuscript.

*Reviewer 4:*

– Experiments done in one figure should support the findings in the other figures. Now all 4 figures are completely separate.

– If you want to draw conclusions about the specific effects of MAIT cells, you need to purify MAIT cells and not total CD8 T cells. i.e. FACS sort CD161+ Va7.2+ cells

– Why study ART at all if you are not going to se this data on a functional level? Remove the data or include it in functional studies.

– Why study PD1 and TIM3 at all if you are not going to look at these markers in the rest of the study? Remove the data or include it in functional studies.

– Why not investigate CCL3/4/5 production by MAIT cells?

---

## [Author Response]

Essential revisions:1. It is well established through multiple studies that MAIT cells (defined by surface marker expression) are depleted during HIV infection and poorly reconstituted during ART. Additionally, changes in GzmB and Tim-3 expression, as well as general MAIT cell activation, during HIV infection have previously been reported (Leeansyah 2013, Leeansyah 2015, and others). It is surprising that the current study does not take advantage of MR1 tetramers to identify the MAIT cell population, as this would present the opportunity to characterize the MAIT population in HIV-infected cohorts in a manner which would be more novel. The data on MAIT cells in the GALT would be stronger if representative staining were shown, but would also benefit immensely from use of the MR1 tetramer to characterize GALT MAIT cells, as this has not previously been done.

We agree that usage of MR1 tetramers would be novel in identifying MAIT cells in GI derived tissue. However, this study commenced prior to these tetramers becoming available for staining, so for consistency we pursued this throughout. We have however recently published an example in the GI tract (Leng et al. Cell Reports, Sept 2019; Supplementary Figure 3C). We have also shown that tetramer positive MAIT cells maintain the CD161++Va7.2+ phenotype in the GALT (please see Author response image 1).

**Author response image 1. sa2fig1:** Double Tetramer positive MAIT cells are CD161++Va7. 2+.

2. Infection of THP-1 monocytes in vitro was used to show that HIV can stimulate MAIT cells via IL-12 and IL-18. The use of THP-1 as the target cells for HIV-1 infections was somewhat intriguing since THP-1 cells usually don't support significant productive HIV-1 infection. With this in mind, it would be better to use cells that support productive HIV-1 BaL infection eg. monocyte-derived macrophages (MDM). It remained unclear are IL-12 and IL-18 produced by infected or bystander THP-1 monocytes? Can inactivated HIV induce IL-12 and IL-18? IL-12/18 produced by THP-1 cells themselves might also have some antiviral effects through various MAIT cell-independent mechanism. Have the authors ruled out this potential effect? Are the levels of IL-12 and IL-18 elevated in HIV+ patients in blood and in tissues?

We used THP1 cells as they are a robust surrogate for monocyte/macrophages and where we have found they respond well to viruses (and bacteria) to drive MAIT cell activation (Van Wilgenburg et al., Nature Comms 2016). We can clarify the background to this experiment further. We turned to this approach, which has been used by many labs now in the MAIT cell field, as the simpler experiment of using ex vivo PBMCs had failed to activate MAIT cells (Ussher et al. Blood 2012). We can however readily test further whether combinations of monocyte-derived macrophages and BAL also work to activate MAIT cells in vitro, as previously tested for the other viruses.

IL-12 and IL-18 production by THP-1 cells is most likely due to activation of these cells by HIV through innate sensing. We have encountered this in previous studies (eg Van Wilgenburg et al., above), where influenza drives MAIT cell activation by the same route, but influenza does not replicate in the cells used. A similar finding using a replication deficient adenovirus in monocytes is also seen and the mechanism has been further explored (Provine et al., BioRxiv 2019). These papers have been referenced (line 290) in order to explain this – overall it seems very plausible that such presenting cells can drive activation even if a full viral life cycle is not completed.

Inactivated HIV did not activate THp1 cells that in-turn did not stimulate MAIT cells to produce IFNγ (Figure 3—figure supplement 1).

This manuscript is focused on the role of IL-12/18 on inducing MAIT cells’ anti-viral function. We show that was no viral inhibition following blockade of CCL3/4/5 with antibodies. IL-12/18 should be in the supernatant, but does not seem to have antiviral effects in this setting.

There are studies where these inflammatory cytokines (and others which may impact on MAIT cell function that have been described) have been studied in acute and chronic HIV infection. We can readily include references to such data in the revised manuscript.

3. The data shown in Figure 3 does clearly demonstrate that exposure of THP-1 cells to HIV results in IL-12/IL-18-driven MAIT cell activation in vitro. It is important to note, however, that this assay recapitulates only a very specific aspect of HIV infection, and does not include any assessment of MAIT cell activation by HIV-infected CD4^+^ T cells or other mechanisms such as microbial translocation. It is surprising that MAIT cell production of IFNγ was not attributed any anti-viral role, considering that similar experiments with iNKT cells have shown that iNKT-derived IFNγ limits HIV infection of primary T cells in vitro (Vasan 2007). Considering recently published data indicating the CCL3 and CCL4 production by MAIT cells is significantly greater following TCR-mediated stimulation than cytokine stimulation (Lamichhane 2019), and the impact that microbial translocation could have on MAIT cell activation, it is unclear what the relative contribution of cytokine-based activation would be on MAIT cells in vivo during HIV infection.

Thank you for the detailed comments. Here we have tried to focus here on the TCR-independent activation of MAIT cells by HIV as this seemed most relevant to their role as antiviral cells – this was prompted by our studies showing such activity is relevant in influenza in vivo. We agree that in the presence of ligand, MAIT cells will be activated via their TCR, and this could certainly affect their overall behaviour in vivo – but this would be a very diffuse effect since the bacteria or ligand would be distributed on non-HIV infected cells. Although this in theory could contribute to MAIT cell activation (in concert with cytokines) and behaviours such as activation induced cell death, this was not what we were originally trying to address in this study. We do have additional data showing the interdependency of TCR stimulation with cytokines, suggesting that even low levels of TCR triggering enhances the response to cytokines. We could include such data to further address that point i.e. that gut translocation could sensitise MAIT cells for antiviral functions.

We would politely disagree with the reviewer about the relative degree of activation with cytokines and the induction of chemokines. 3 papers were published in parallel recently which included the Lamichhane paper. In our own paper where we compare cytokine induced vs TCR triggered MAIT cell activation and we see a much larger response with cytokines and where clear activation of the relevant chemokines is seen (Leng et al., Cell Reports, Sept 2019; Figure 4F and G). The differences in the papers are due to slightly different timings and stimulation conditions – overall they are relatively congruent as we tested in our study through data integration, as well as similar to in vitro and in vivo mouse data. Indeed, generally all groups have found that cytokine induced activation is slower but more sustained. Certain functions of MAIT cells are highly TCR dependent (even if amplified by cytokines), but we do have good data that the antiviral chemokines are readily induced by cytokines.

Regarding the IFNγ, we can certainly test this using blockade as we have done in other settings (e.g. HCV in vitro and influenza in vivo). However, we note that blockade of chemokines neutralises the antiviral effect completely, so unless both are required, we think the impact in our assay is likely to be small. We have added this in the Discussion section (Line 323-328)

4. Did the authors also see contact-dependent MAIT cell-mediated inhibition of HIV replication? Similarly, would TCR-activated MAIT cells mediate antiviral activity through CCL3-5 production and/or contact-dependent mechanisms?

We did not observe any contact-dependent inhibition of HIV (Figure 4—figure supplement 2)

5. Technically, there is no indication of the use of uninfected THP-1 cultures as a control in Figure 3A or B, and it is unclear as to whether the data presented in Figure 3B is background subtracted. Sorting on CD8^+^ T cells prior to culture with the THP-1 cells indicates that only CD8^+^ MAIT cells were studied in this experiment, which is a caveat that should be noted, as MAIT cells can include a substantial CD4-CD8- population as well.

The figure legend has been amended to indicate the use of uninfected THP-1 cells as controls (Figure 3B). A caveat has been included in the Results section to explain that MAIT cells can also be found in the CD4-8- subset (Line 157-159).

6. The assay in Figure 4 using the CEM-GXR cells is likely biased to detect an impact of β-chemokines on HIV entry, given the cell line overexpresses CCR5. However, it would be important to confirm MAIT cell-mediated anti-HIV activity using primary human immune cells that productively support HIV-1 BaL replication, such as MDM and/or PHA/IL-2-activated PBMCs, plus measurements of productive viral replication (eg. p24, RT activity, etc).

HIV inhibition by MAIT cells was also observed in Primary PBMCs (Figure 4B). Here we used HIV-iGFP (JR-FL) R5 tropic virus.

7. Have the authors tested other R5 strains apart from BaL? It would be important to see that this antiviral effect is not just restricted to the BaL strain of R5 HIV-1. Along this line, there were only n=4 of presumably independent MAIT cell donors in Figure 4B (or was it 4 different CEM-GXR expts but using a single or pooled MAIT cell donors?). It would be good to increase the sample size to strengthen the manuscript. Similarly, have the authors tested X4 strains, lab-adapted or primary isolates? It would be an important control and interesting to see if MAIT cells can also/cannot inhibit X4 viruses. Along this line, did MAIT cells produce CXCR4-binding chemokines (eg. CXCL12/SDF-1)?

We have also used HIV-iGFP (JR-FL) R5 tropic virus. The total n for donors in these experiments = 8. These were 8 individual donor supernatants and not pooled. We have not tested X4 strains, as expression CXCL12 from MAIT cells were very low (Figure 3—figure supplement 3).

8. The authors show that IL-12 + IL-18 stimulation induces apoptosis in MAIT cells. While this is interesting observation, it does not explain why there is no increase in MAIT cell numbers following years of ART treatment. One can hypothesize that due to lower HIV load there will be less of IL-12 and IL-18, and subsequently less of MAIT cell stimulation. This is the case with the expression of GzmB, PD-1 and Tim-3. However, the numbers of MAIT cells remain the same. This discrepancy should be commented.

This is an interesting point for discussion. There are many unknown issues regarding MAIT cell (re)generation in adults. It takes around a decade to expand the MAIT cell population in early life, and the cells start to decline with age in a very reproducible way. These features suggest that the turnover of this subset is different to regular CD8^+^ T cell memory cells and so they may overall recover slowly following a depletion. A major caveat to this is that such studies have been largely performed in blood. Thus the overall recovery may not be solely viral load dependent but depends on features such as age, cell turnover and cell redistribution. We have submitted a proposal to study MAIT cell turnover (in heath and HIV infection) using heavy water but this is out with the scope of this project. Discussed in lines (302-307).

9. The methods section lacks important details for several experiments. The *E. coli* stimulations do not appear to be described at all. There is no indication of the number or ratio of T cells used in the THP-1 co-culture experiments.

The methods section has been amended to be more detailed (line 460-480)

[Editors' note: further revisions were suggested prior to acceptance, as described below.]

Below please find the summary of major concerns raised by reviewers.1) The data in Figures 1 and 2 remain to be only confirmatory of multiple previously published studies. Without the data on GALT cells studied with MR1 tetramer, this portion of the manuscript does not add new information to the field. The authors argument that "this study commenced prior to these tetramers becoming available for staining, so for consistency we pursued this throughout" is somewhat in contradiction with their sentence stating that they have recently published the paper using MR1 tetramers to demonstrate MAIT cells in a GI tract. Some attempt at providing a confirmation of data should be done.

Figures 1 and 2 were used to show the depletion of MAIT cells in blood and tissue during HIV infection. Although this does initially confirm other studies, here we use patient samples form primary and chronic infection, as well as, in particular, a set of very difficult to obtain long-term non-progressor/Elite controller samples. We have now in Figure 2—figure supplement 1- added a dataset to clearly show that expression of CD161 and Vα7.2 is comparable to MR1-5OPRU tetramer staining in gut tissue for the identification of MAIT cells. This data is quite robust as we used a specific dual-staining approach and a good negative control with careful gating.

Tetramers were not available at the time for staining of HIV+ patient tissue samples – these samples were obtained from the HEATHER cohort and samples were stained real-time during collection. The data on MAIT cells in the GI tract – taken at the time of an operation for cancer – were obtained after the HIV study. The data obtained here showing a side by side comparison are very useful for the field and hopefully displayed in a clear format that allows for evaluation by the reviewers and ultimately the readers.

This paragraph has been added to the Results section.

“While MR1-Tetramers loaded with the MAIT cell ligand 5OPRU represent the most reliable way to identify MAIT cells in the blood, MAITs can also be identified as CD161++ Vα7.2+ cells, especially within the CD8^+^ population (Kurioka et al., 2017). In order to test whether this would also be the case in the gut, we co-stained CD161 and Vα7.2 with 5-OPRU-loaded or control MR1-tetramers and assessed whether CD161++ Vα7.2+ cells stained positive for the tetramers (Figure 2—figure supplement 1). Within the CD8^+^ and double-negative (DN) populations, almost all CD161++ Vα7.2+ cells reacted with both 5-OPRU-loaded but not the 6FP-loaded control tetramers, suggesting that the combination of high CD161 expression with Vα7.2 can be used to identify MAIT cells within the CD8 and DN T-cell populations in the gut. CD161++ Vα7.2+ expression on CD8^+^ T-cells was comparable with MR1-5OPRU- tetramer+ cells and could identify the majority of MAIT cells within this compartment.”

2) The response to reviewer’s suggestion to assess the cells that support productive HIV-1 infection to stimulate MAIT cells instead of THP-1 cells, was also not clear. The authors state: “We can clarify the background to this experiment further” or “We can however readily test further whether combinations of monocyte-derived macrophages and BAL also work to activate MAIT cells in vitro, as previously tested for the other viruses …”. Still, it remained unclear whether the authors performed the requested experiment and if so, it should be included in the manuscript. Since inactivated HIV did not result in THP-1 activation of MAIT cells, the question is what mechanism is behind this finding. This was not further discussed in the manuscript.

We have now had the chance to address this directly by using unmanipulated human PBMCs as targets and infecting with a GFP expressing JRFL virus. This is now shown in figure 4B. We were able to reproduce the data previously obtained using THP1 cells. IL-12/18 stimulated MAIT cell supernatant was able to inhibit infection of PBMCs in vitro. We hope this use of primary cells explores the area suitably for the reviewers and addresses the question posed.

We have included more data on HIV inactivation. This is now discussed in a section (line 279-289) on inactivated HIV as outline below.

“The THP1 monocytic cell line was used as it has been extensively trialed by our group and many others as an effective APC for MAIT cells in microbial and viral infections (Ussher et al., 2018, van Wilgenburg et al., 2016). MAIT cell activation was dependent on IL-12 and IL-18 produced by THP1 cells via innate sensing (van Wilgenburg et al., 2016a, Provine N, 2019). THP1 sensing of HIV-BaL may be through TLR7/8 or cytosolic RIG-like receptors (Diget et al., 2013, Guo et al., 2014), which in-turn activates THP1 cells to secrete inflammatory cytokines such as IL-12 and IL-18 (Bandera et al., 2018). Inactivated virus was incapable of activating THP1 cells. This may be due to cross-linking of nucleocapsid P7 protein by aldrithiol-2, which may not allow single stranded HIV RNA within virions to bind to TLR7/8 (Rosio et al, 1998).We note that inactivation of influenza and HCV also impacted on MAIT cell recognition in vitro (van Wilgenburg et al 2018), even in a macrophage culture where true infection and replication does not occur. ”

3) Potential anti-viral role of IFNγ was not explored. The authors state: "Regarding the IFNγ, we can certainly test this using blockade as we have done in other settings (eg HCV in vitro and influenza in vivo)."; Again, it is unclear whether or not these experiments have been performed. Although we agree that it is unlikely that IFNγ would result in a block of viral entry, it would still be relevant to test the impact of other MAIT cell-secreted cytokines and chemokines on viral replication, as the presence of MAIT cells at mucosal surfaces is clearly insufficient to block HIV infection. If the authors believe that mechanisms are the same in case of other viruses, we do not see the point of this study.

Apologies for this confusion but the original statement was from the first response to the editor and carried over. We have now had the chance to address this experimentally. We found that IFNg did not have anti-HIV effects when blocking antibodies were used. This is now Fig.4-figure supplement 4. Low antiviral effect of IFNg has been documented by other groups. This has now been added to results section (line 221-225) and the discussion section (line 277-279 and 314-324).

“IFNỿ expression in MAIT cells was observed when HIV-BaL was used to stimulate the antigen presenting cell (THP1), but had no anti-HIV effect.”

“Both MAIT cells and THP-1 cells will produce a broad range of factors following stimulation as recently described (Hinks et al., 2019, Lamichhane et al., 2019, Leng et al., 2019). However, even though factors such as IFNỿ could potentially possess anti-HIV activity, as has been described for iNKT cells (Paquin-Proulx et al., 2016), we were able to block the suppressive activity very effectively with anti-chemokine antibodies, suggesting that at least in our models, these are the most potent effectors. There has been no significant difference in HIV specific IFNỿ response reported in both progressors and long-term non-progressor patients with chronic HIV infection (Roff et al., 2013, Zanussi et al., 1996). This is consistent with original in vitro studies which revealed no antiviral effect of IFNg and even enhancement of infection in primary cells and several clinical trials which revealed no impact of this cytokine in vivo.”

4) Control experiments using X4 viruses will further strengthen the manuscript, given the inhibition by MAIT cells is acting on R5 viruses (and thus no inhibition should be seen on X4 viruses).

We have shown MAIT cells to express low levels of the CXCR4 ligand (Fig. 3-figure supplement 3), much lower than beta chemokines in our assays. We have now had the opportunity to address the experiment suggested and show that anti-HIV effect of MAIT cells is confined to R5 tropic viruses. We used HIV-LAI (X4 tropic) virus and saw no inhibition of infection when stimulated supernatant from MAIT cells were added (Fig.4 figure supplement 2), highlighted in lines 212-214.

[Editors' note: further revisions were suggested prior to acceptance, as described below.]

Reviewer 4:– Experiments done in one figure should support the findings in the other figures. Now all 4 figures are completely separate.

Thank you for the comment. We think this is a critique of the flow of the paper. The idea behind it is as follows: 1. Show MAITs are activated in vivo (here the addition of ART makes sense as it shows the role of viral replication and acts an in vivo control). 2. Address the mechanism in vitro. 3. Show the activation leads to relevant functional consequences for the MAIT cell. 4. Show that this responsiveness is meaningful in terms of antiviral responses against HIV. We think this combination of data, presented in this order – and adjusted to incorporate all the previous reviewers’ comments – hangs together reasonably. If we just went for the in vitro mechanisms we would not really be able to link this to relevant in vivo findings and vice versa, so the approach demands a bit of both. It follows a similar flow to a previous paper where we studied HCV, and which had a different mechanism (van Wilgenburg et al., Nat Comms 2016). We did need to address the issue of cell redistribution as this is unclear currently and actually highly relevant to the antiviral role.

We have tried to smooth all of this out throughout, so it is a bit easier to follow. We also reordered the last part of the paper (Figure 3 and 4; lines 201-258) to help with this. New amendments are highlighted within the revised manuscript (related manuscript file).

– If you want to draw conclusions about the specific effects of MAIT cells, you need to purify MAIT cells and not total CD8 T cells. i.e. FACS sort CD161+ Va7.2+ cells

We have now added new experimental data that was performed on FACS sorted MAIT cells and our original data on bulk CD8 T cells were reproducible (Figure 3C). We show that the major producer of CCL4 are indeed CD161+ Va7.2+ MAIT cells (Figure 4C and D). This is pretty much consistent with all our prior data (and that of other groups) showing the very high responsiveness compared to other T cells of human MAIT cells in the presence of such cytokine combinations.

– Why study ART at all if you are not going to se this data on a functional level? Remove the data or include it in functional studies.

Patient data following ART was to show the impact on activation and exhaustion of MAIT cells that is highly evident during HIV infection. We showed on figure 2b that even during acute HIV infection MAIT cell frequencies are dramatically reduced compared to healthy controls. There is also very high expression of PD-1, Tim-3 and GzmB in these cells ex vivo. This response is modified by ART at the level of phenotype as clearly shown here. Any functional work on these cells is technically very difficult as they die in culture. Hence, we sorted MAITs from healthy donors and show anti-HIV function when they are activated by the virus, or with IL12 and IL-18 as a reduced surrogate. This is for proof of principle. Thus, we were not trying to address specifically the impact of chronic stimulation or ART on MAIT cell function – we needed to prove first whether this is of any impact for HIV. If MAIT cells are diminished for any reason (by deletion is the most obvious) then this effect will be reduced.

– Why study PD1 and TIM3 at all if you are not going to look at these markers in the rest of the study? Remove the data or include it in functional studies.

The addition of these markers was to assess the impact of HIV on MAIT cell activation. We could have used a range of alternatives – for example CD69 as we have used elsewhere but these markers show a good dynamic range and staining is well established. We also used GzmB which we know is highly responsive (see eg van Wilgenburg paper as quoted earlier). Having shown this responsiveness in a robust way, we don’t think using these markers in later studies is needed to address the point we were trying to understand, which is whether this activation is functionally relevant. For this reason, we have left it in but described the meaning in more detail.

– Why not investigate CCL3/4/5 production by MAIT cells?

This is now included in Figure 4A, 4B, 4C and 4D.